# Quantifying the Long-Term Performance of Rainwater Harvesting in Cyclades, Greece

Ioannis Zarikos *, Nadia Politi, Nikolaos Gounaris, Stelios Karozis, Diamando Vlachogiannis and Athanasios Sfetsos

Environmental Research Laboratory, National Centre for Scientific Research "Demokritos", 15310 Athens, Greece; nadiapol@ipta.demokritos.gr (N.P.); skarozis@ipta.demokritos.gr (S.K.); mandy@ipta.demokritos.gr (D.V.)
* Correspondence: i.zarikos@ipta.demokritos.gr

**Abstract:** South European and Mediterranean countries traditionally suffer from water scarcity, especially the regions around the Mediterranean. In Cyclades, the effects of drought have historically been observed and tackled with small-scale applications, with the most efficient method being rainwater harvesting (RWH). RWH is an inherent aspect of the local population's culture and architecture, since most houses have built-in water tanks and flat roofs to harvest as much rainwater as possible. In recent decades, the increase in local population and tourism have added additional stress to the limited water resources of the Cycladic islands. To overcome water shortages, most of the islands are equipped with desalination plants. Despite the use of these plants, RWH is still a vital source of water that is free and has zero carbon footprint. Thus, it is important to compare, assess and quantify the performance of this traditional water conserving method as a key water source for the islands' water resources management, today and for the coming decades. In this research, we investigate and quantify the future performance of rainwater harvesting applications and their contribution to continuous, sustainable, and climate-resilient water supply. The results show a decrease in rainwater harvesting potential in most of the islands, as well as the negative effect of touristic activity on per capita water availability on the islands.

**Keywords:** climate change; water scarcity; rainwater harvesting; drought; climatic simulations

## 1. Introduction

The Mediterranean region is known for its unique climate, characterized by long, hot, and dry summers, and mild, rainy winters. Water scarcity is a significant challenge in the Mediterranean region due to the limited availability of water resources and increasing demand for water due to population growth, tourism, urbanization, and agricultural development.

According to the Cross-Chapter Paper report for the Mediterranean Region (CCP4) [1], due to climate change an increase in hazards is expected, such as heat stress, storms, extreme rainfall, internal and coastal flooding, landslides, air pollution, drought, water shortages, sea level rise, and storm surges. In the report, drought and water scarcity are identified as the main hazard in the region. Rural areas are expected to experience a major impact on water supply, food security, infrastructure, and agricultural incomes, as well as a shift in food and non-food crop production areas. The global trend for drought is not well predicted due to a lack of direct observations, dependencies on the selection of a definition of drought, and geographical inconsistencies in the trend for drought. Since the mid-20th century, the worldwide assessment of drought change has been influenced by low confidence; observational uncertainties and difficulties in identifying the variability of drought at the end of the decade and long-term trends have also been observed. According to regional observations, most droughts and extreme rainfall events of the 1990s and 2000s have been the worst since the 1950s [2], and certain trends in total and extreme precipitation amounts are observed (WGI AR5). In the Mediterranean region, according to CCP4 [1],

the expected frequency and severity of droughts are projected to increase (severe emission scenarios). In addition to that, the overall trends in the hydrologic cycle indicate a decrease in river runoff, groundwater recharge, and water availability in rivers and lakes [3–5]. Water scarcity and drought in the Mediterranean countries are the phenomena that have historically troubled the local population. Due to climate change and over-population in these regions, the negative effect of these phenomena has been more profound in recent decades. In the literature, it is emphasized that changes in drought frequency and increase in population put domestic water supply at risk [6]. According to the research carried out in the Western Mediterranean region [7], below-average precipitation for the studied years indicated a decrease in available water resources and the drought of 2022. Climatic simulations for this region show that water scarcity in the studied region and the associated water quality decrease will be more profound in the future, based on RCP 4.5 and 8.5 scenarios [8]. The same observation was also carried out for Eastern Mediterranean, but with lesser negative effects, for the specific period. It is important to note that Northern European countries, traditionally not affected by water scarcity, have also now started to experience water scarcity and drought events.

Traditionally, to battle water scarcity, communities in the Mediterranean countries employed rainwater harvesting techniques (RWH) [9] to collect rainwater, primarily for drinking and secondarily for irrigation purposes; this practice is carried out even at present. Rainwater harvesting is a technique used to collect and store rainwater for later use, especially in areas where water scarcity is a concern. The technique involves capturing rainwater from rooftops, roads, and other surfaces, and storing it in tanks or underground reservoirs for later use. Rainwater harvesting has several advantages in addressing water scarcity [9–13]. As it involves capturing and storing rainwater, it thus conserves water resources by reducing the dependence on surface and groundwater sources, which are often overused and depleted. Moreover, it is a cost-effective way to obtain water, as it eliminates the need for expensive water treatment and production processes (purification, desalination, etc.) and reduces the energy costs and carbon footprint associated with them. In addition to that, it is considered to improve local water self-reliance since by collecting and using rainwater locally, communities can become more self-reliant in meeting their water needs, especially in areas where the water supply is unreliable or inadequate. Overall, rainwater harvesting is an effective technique for addressing water scarcity, and it can be implemented at different scales, from individual households to large commercial and industrial complexes. However, it requires careful planning and management to ensure that the collected water is safe and suitable for its intended use.

However, due to climate change, overpopulation, and tourism, water resources in these regions are overstretched, with RWH applications being considered as obsolete and occasionally ineffective to meet water demands, with damaging effects in most cases (i.e., aquifer salinization, decreased water resource availability, etc.) [1,14,15]. Hence, alternative water resources (i.e., brackish or seawater desalination and storm-water collection), which complement the existing water resources, are used to cover the increasing water demands. Despite that, RWH is still a vital water source for meeting a major percentage of water demands in the region.

The existing RWH applications can be divided into two major categories based on their scale and construction fundamentals: rooftop [16–19] or land-based applications [20–22]. Rooftop RWH applications take advantage of the roof area on the buildings, either flat or inclined roofs, and using a pipe network transfer that water into water tanks (attached on or not to the building) for storage. The harvested water is mainly used for each specific building. On the contrary, land-based RWH applications are larger in scale and are used to supply water to multiple users. The performance of a rainwater harvesting system depends on various factors such as the amount of rainfall, the size and design of the system, the quality of the water collected, and the demand for water. The effectiveness of RWH has been the subject of numerous research works.

It is common practice, as well as more accurate, to analyze the effectiveness of RWH applications based on historical and measured data [16,17,23,24]. However, it is important that such an approach is complemented using near and far future climatic simulation results to better assess the performance of RWH. The future effectiveness of RWH applications analysis, based on climatic simulations, has been reported in recent works [18,21]. Islam et al. (2021) researched a relatively small study area, covering 109 RWH rooftop applications. Adham et al. (2019) [21] investigated a larger area, focusing solely on land-based RWH applications. At a similar scale, the negative effect of climate change has been investigated for rooftop applications in Adelaide, Australia [25]. Finally, an insightful global RWH analysis [26] demonstrates the paramount importance of RWH applications in a variety of climatic conditions. Despite the wealth of information on the effectiveness of RWH applications worldwide, there is still limited data on the efficiency of the existing RWH system in the Mediterranean region.

In the Cyclades, Greece, the effects of drought have historically been observed and tackled with small-scale applications, with the most efficient method being rainwater harvesting (RWH). RWH is an inherent aspect of the local population's culture and architecture as most houses have built-in water tanks and flat roofs to harvest as much rainwater as possible. Over the past decades, the increase in local population and tourism have added additional stress to the limited water resources of the Cycladic islands. To overcome water shortages, most of the islands are equipped with desalination plants and in some cases, the groundwater reserves have been overexploited. Despite the use of these water resources, RWH is still a vital source of water that is freely available and has zero carbon footprint. Thus, it is important to compare, assess, and quantify the performance of RWH as it is considered the primary source of drinking water and an alternative water resource (AWR) for irrigation purposes. This paper addresses for the first time the long-term performance of traditional rainwater harvesting applications in the South Aegean region, more specifically in the Cyclades island complex. It presents a novel approach which utilizes spatial analysis in combination with climatic model predictions, allowing for the future evaluation and quantification of the RWH performance indicators in near and far future, in comparison with historical data. In addition, emphasis is laid on the negative effect of tourism and seasonal population variations, which have not been previously taken into account in the literature; however, it has been proven that these factors have a dramatic effect on water resources availability. Such indicators will be valuable in the development of an up-to-date integrated water resource management (IRWM) system in the Cyclades, which can also be employed to other Mediterranean regions that are on the lookout for alternative sustainable water resources.

## 2. Materials and Methods

### 2.1. Study Area

The study area is the Cyclades Island complex (Figure 1), which is part of the South Aegean region, consisting of 24 inhabited islands, out of the total 220 islands and skerries. The total area of the island complex is 2572 km$^2$, representing 2% of Greece's total area, and it has a population of 119,549 people, based on the latest population census of 2021. Given the tourist activity in this region, especially in the summer months, the population increases dramatically. In 2021, more than 6,300,000 tourists visited the Cycladic islands, with an average stay of 5 days. This periodic population variation has a direct impact on the water reserves of the islands, which are already scarce. Historical precipitation data (1980–2004) indicate that the precipitation in the Cycladic islands averages around 300 mm/year. However, model predictions show a decreasing trend in the average precipitation, both RCP 4.5 and RPC 8.5. The predicted decrease in precipitation is around 10% (RCP 4.5) and 20% (RCP 8.5) for the period of 2025–2049 and 10% (RCP4.5) and 30% (RCP8.5) for the period of 2075–2099 [27].

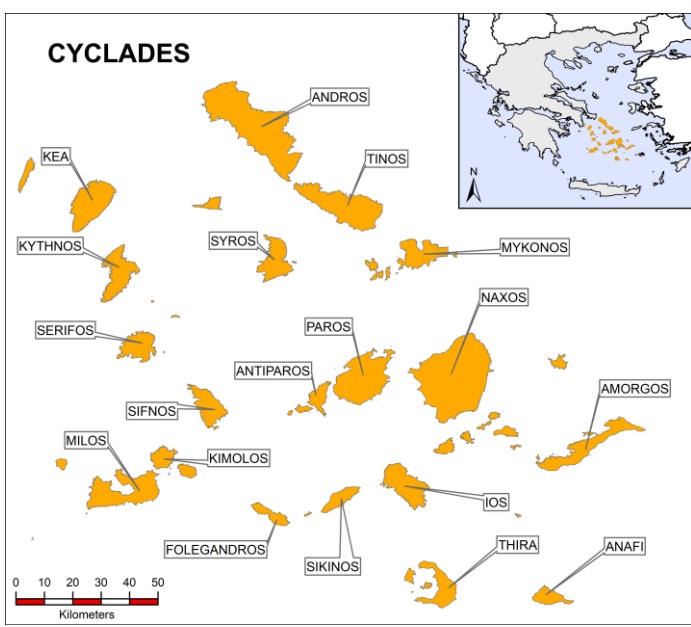

**Figure 1.** Study area locations.

### 2.2. Climatic Model

High-resolution climate simulation precipitation datasets of 5 km horizontal resolution for the area of Greece were derived via the dynamical downscaling technique using the non-hydrostatic Weather Research and Forecasting model (WRF/ARW, v3.6.1) [28] in a one-way nesting setup composed of two domains. The first domain (d01) of 20 km resolution is centered in the Mediterranean basin at 42.5 N and 16.00 E, and a second nested domain of 5 km is centered in Greece. The large-scale circulation in the 20 km domain is nudged towards the forcing boundary conditions with spectral nudging. The model setup has the following physics options: the Mellor–Yamada–Janjic scheme (MYJ) [29] for the planetary boundary layer (PBL) scheme, the WRF single-moment six-class scheme (WSM-6) for cloud microphysics and Betts–Miller–Janjic' scheme for the cumulus parameterization. The Rapid Radiative Transfer Model, RRTM, for both longwave and shortwave radiation and finally, the Noah LSM was employed as the land surface model. The initial and boundary conditions for the climate change assessment are derived from EC-EARTH climate simulations for RCP 4.5 and RCP 8.5 scenarios and encompass time slices representative of the historical or reference (1980–2004), near future (2025–2049), and far future (2075–2099) periods. For future projections, the equivalent $CO_2$ concentration was updated every year according to the emission scenario in the WRF simulations. Climate simulation data and model setup are based on a number of extensive attentive validation studies of the application of the regional model (WRF) with the reanalysis datasets of ERA-Interim [30–32] and the GCM (EC-EARTH) [27,33,34]. Those studies proved the capability of the downscaling process in capturing the spatial and temporal climate patterns of precipitation, temperature, and wind speed for Greece by comparing the downscaled historical simulations with the available meteorological data from the Hellenic National Meteorological Service (HNMS). Moreover, for this study, projected changes in drought severity and drought duration were illustrated to emphasize the increasing drought conditions in the vulnerable areas of the Cyclades due to climate change. Duration and severity were estimated through the calculation of 12-month Standardized Precipitation Index (SPI). These elements, known as drought characteristics, are a subset of datasets for the area of Greece estimated in the work by Politi et.al (2022) [32], who investigated the projected changes in drought characteristics in the same high resolution for Greece. Therefore, further detailed information on the methodology of the calculation for drought index and drought characteristics can be found in the related study.

### 2.3. Data Description and Processing

This study utilizes two groups of datasets based on the nature of climatic data (precipitation): historical data and future predictions. The historical data cover the period between 1980 and 2004, while the future prediction data cover the periods of 2025–2049 and 2075–2099. Each dataset is linked to the population census of the investigated period, 1991 for the historical data and 2021 for the predicted data.

As shown in Figure 2, the climatic data in each domain are linked to the corresponding RWH applications (flat roofs or water catchments) in that domain and the islands hosting these applications. The total annual harvested volume of each island is calculated and divided by the total population to measure the per capita harvested rainwater in each island, which indicates the RWH potential of each island.

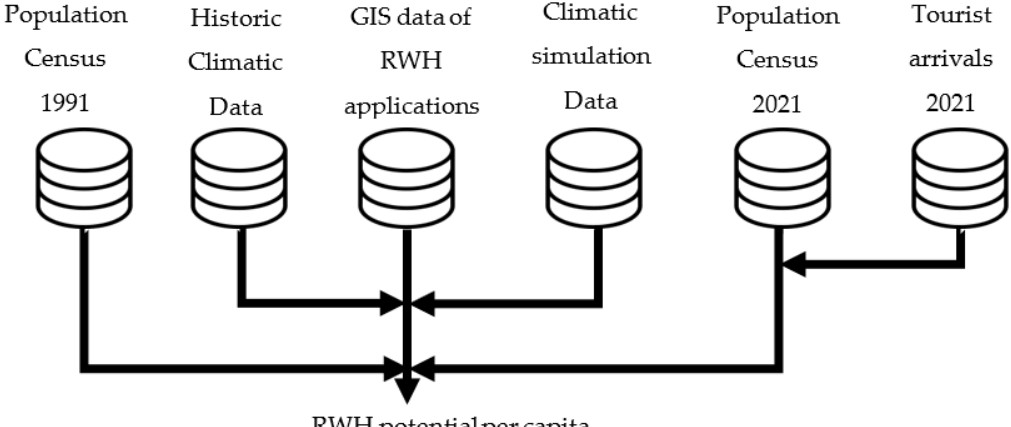

**Figure 2.** Data analysis workflow.

The effect of tourism on RWH potential has also been assessed based on tourist arrival data. Given the limited availability of data, only the most touristic islands were included in the analysis. The normalized touristic population that can be added as potential local population was calculated using Equation (1):

$$T_n = \frac{s}{T_{tot}} \tag{1}$$

where $T_n$ is the normalized touristic population, $s$ is the number of days that tourist stayed on the island, and $T_{tot}$ is the total number of tourists who visited the islands over the duration of one year.

### 2.4. Water Scarcity Index

In the literature, there are multiple water scarcity/stress indexes that can be calculated based on different approaches and target areas [35]. In this research, we use two water stress index approaches, Falkenmark Water Stress Indicator (FWSI) [36] and Basic Human Water requirement [37], which aim to describe the regional extent of water scarcity and the extent of human activities and needs, respectively. FWSI is one of the extensively used water scarcity indicators. This indicator considers only the available renewable water (rainfall not returned to the atmosphere via evaporation and evapotranspiration) per capita per year. FWSI defines a threshold of 1700 $m^3/p \cdot$ year, below which the region is considered to experience water stress. As the water availability decreases further from this threshold, the negative effects become more severe. More specifically, water availability below 1000 $m^3/p \cdot$ year is considered a limitation for economic development, while 500 $m^3/p \cdot$ year is considered a life constraint.

Basic Human Water requirement (BHWR) [37] defines the minimum requirement for daily water use per capita per day as 0.05 $m^3$ (50 L/p $\cdot$ day) in order to cover the basic needs of each person corresponding to 18.25 $m^3/p \cdot$ year. However, given the need of every

individual, this minimum water requirement can reach up to 70 L/p · day, a total of 25.5 m$^3$/p · year (Table 1).

**Table 1.** Water requirement for the basic needs per capita/year.

| Usage | Amount of Water Needed (L/p · year) |
|---|---|
| Drinking water | 2–4.5 |
| Sanitation | 20 |
| Bathing | 15–25 |
| Food preparation | 10–20 |

## 3. Results

### 3.1. Drought Duration and Severity

Increasing values of duration and severity are observed during both the periods and climate change scenarios in the study area of Cyclades, indicating that almost all islands will face prolonged and severe drought events in the near and far future (Figures 3 and 4) [32].

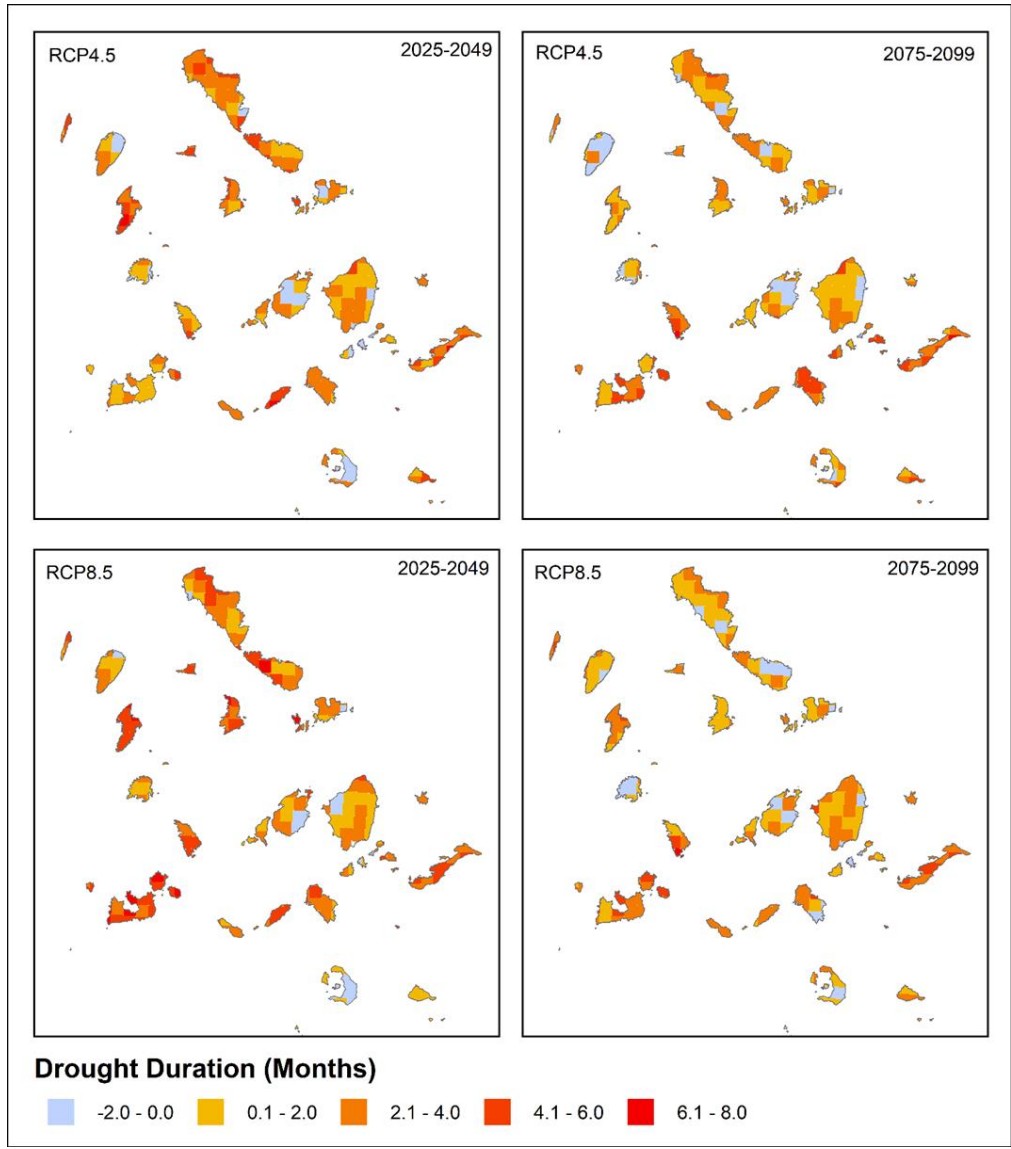

**Figure 3.** Drought duration (months) based on RCP 4.5 scenario for 2025-2049 and 2075–2099 (**top**) and RCP 8.5 for 2025–2049 and 2075–2099 (**bottom**).

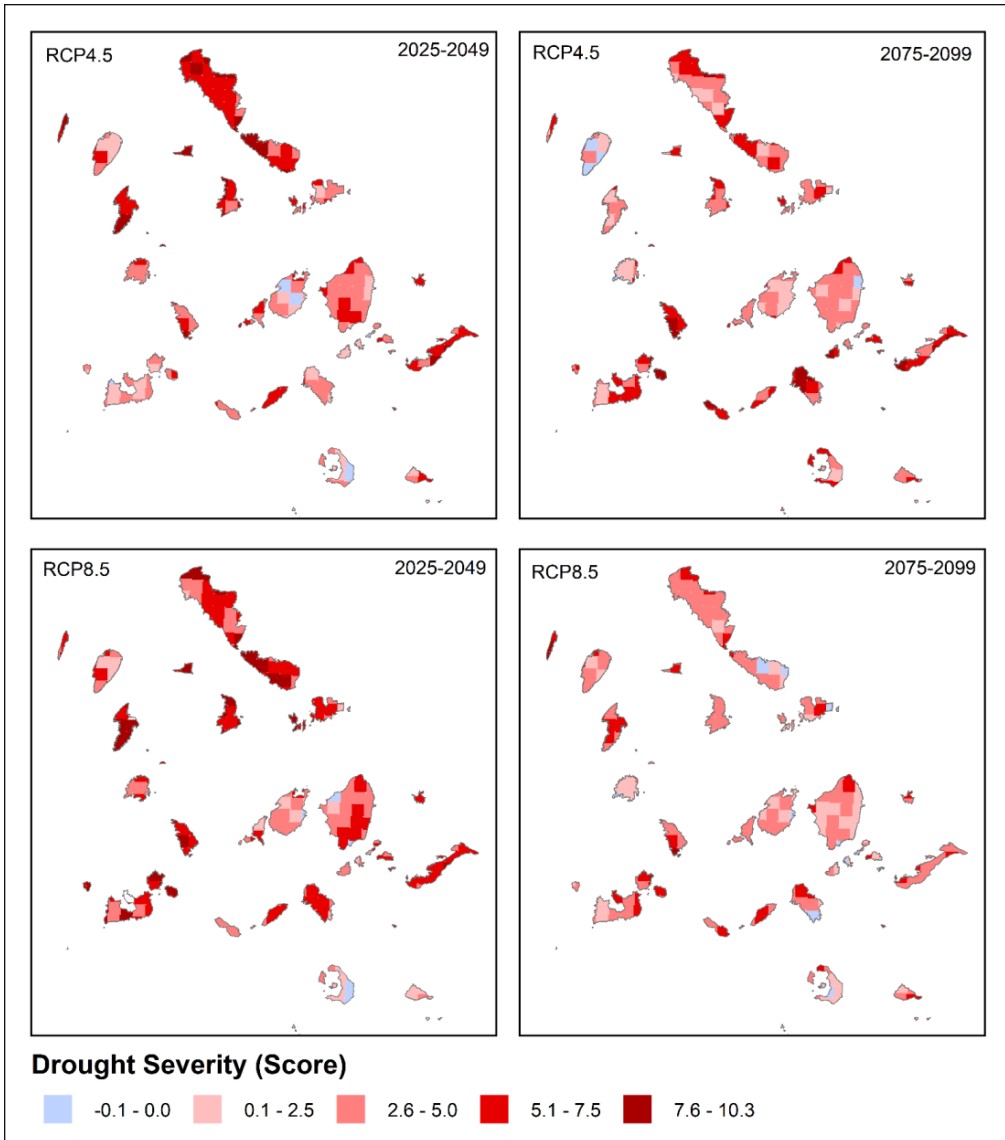

**Figure 4.** Drought severity (score) based on RCP 4.5 scenario for 2025–2049 and 2075–2099 (**top**) and RCP 8.5 for 2025–2049 and 2075–2099 (**bottom**).

Drought duration in the Cycladic islands, with an exception of some small areas, is expected to last longer than 2 months, with a maximum duration of 8 months. As shown in Figure 3, for the near future scenario (2025–2049), the islands of Kea, Tinos, and Paros are the least affected islands with short drought durations. On the contrary, the islands of Kythnos, Ios, Amorgos, Sikinos, and Syros will continue to suffer from prolonged drought events lasting from 4 to 8 months. Such observations are evident in the recent literature [32,38], more specifically in the literature on drought events lasting longer than 7 months. The observations from far future scenarios (2075–2099), regardless of the scenario, indicate a decrease in drought events duration, even for the most affected areas.

Similar trends are observed for the severity of the drought events in the study area. Near future scenarios (2025–2049), presented in Figure 4, show that the islands of Andros, northern Tinos, and western Cyclades islands will be affected by most severe drought events. On the contrary, eastern Cyclades will experience less severe drought events, except for Amorgos. Drought severity is projected to ease in the far future scenarios (2075–2099), with RCP 8.5 showing the most optimistic results. Kythnos and Amorgos are the islands

with the most severe drought events. The rest of the islands will experience moderate drought events.

### 3.2. Regional Rainwater Harvesting Performance

We define regional RWH potential as the total amount of rainwater per capita that can be collected and stored, either in water tanks or surface water reservoirs. As previously noted, in the RWH system, the flat rooftops of the buildings have traditionally been used as the main tool for collecting rainwater. Stormwater harvesting applications have also been employed in specific locations. Although such applications are not within the scope of this research, they are a crucial alternative water resource that can contribute significantly to the RWH volume of the Cycladic islands.

The RWH potential in this region is directly related to the rooftop area and water catchment coverage in relation to the total area of the islands. Table 2 shows the percentage of RWH application, both rooftop and land-based, in relation to the total surface area of the islands. The mean precipitation for all three 25-year periods have been used the average precipitation in this research. Despite the small area coverage of the RWH applications, which is below 10% in all the islands, the harvested rainwater has a significant volume that can partially cover the water needs in these islands.

**Table 2.** Volume of harvested rainwater vs. the total volume of rainwater precipitated on each island.

| Island | Total RWH Area (×1000 m²) | Total Island Area (×1000 m²) | Average Precipitation (m) | Total Volume of RWH (×1000 m³) | RWH (×1000 m³) | Rainwater Harvested (%) |
|---|---|---|---|---|---|---|
| Amorgos | 450.24 | 119,780.99 | 0.41 | 48,738.78 | 183.20 | 0.38% |
| Anafi | 72.95 | 38,170.89 | 0.11 | 4040.88 | 7.72 | 0.19% |
| Andros | 2286.06 | 378,769.54 | 0.89 | 337,781.03 | 2038.67 | 0.60% |
| Antiparos | 521.97 | 34,576.11 | 0.18 | 6180.30 | 93.30 | 1.51% |
| Donousa | 54.92 | 13,398.99 | 0.06 | 835.18 | 3.42 | 0.41% |
| Folegandros | 212.57 | 32,206.19 | 0.11 | 3692.58 | 24.37 | 0.66% |
| Ios | 506.62 | 108,571.87 | 0.32 | 34,205.69 | 159.61 | 0.47% |
| Iraklia | 46.45 | 18,139.70 | 0.08 | 1519.69 | 3.89 | 0.26% |
| Kea | 1035.82 | 131,706.72 | 0.28 | 37,196.30 | 292.53 | 0.79% |
| Kimolos | 197.28 | 37,490.25 | 0.14 | 5277.96 | 27.77 | 0.53% |
| Kythnos | 554.14 | 98,946.34 | 0.23 | 23,080.26 | 129.26 | 0.56% |
| Milos | 1282.98 | 157,628.99 | 0.44 | 69,468.68 | 565.42 | 0.81% |
| Mykonos | 3956.69 | 86,366.54 | 0.34 | 28,994.62 | 1328.32 | 4.58% |
| Naxos | 3375.22 | 428,632.97 | 0.83 | 356,915.77 | 2810.50 | 0.79% |
| Pano Koufonisi | 123.88 | 5712.14 | 0.06 | 337.96 | 7.33 | 2.17% |
| Paros | 4573.30 | 195,332.09 | 0.51 | 99,969.01 | 2340.57 | 2.34% |
| Schinousa | 71.70 | 8093.64 | 0.08 | 678.21 | 6.01 | 0.89% |
| Serifos | 478.86 | 72,396.06 | 0.29 | 21,159.41 | 139.96 | 0.66% |
| Sifnos | 944.19 | 76,225.17 | 0.25 | 18,755.46 | 232.32 | 1.24% |
| Sikinos | 79.44 | 41,316.10 | 0.12 | 4768.23 | 9.17 | 0.19% |
| Syros | 3083.19 | 84,262.41 | 0.26 | 22,021.73 | 805.78 | 3.66% |
| Thira | 6519.71 | 75,966.54 | 0.26 | 19,886.74 | 1706.75 | 8.58% |
| Thirasia | 98.31 | 9163.80 | 0.08 | 734.41 | 7.88 | 1.07% |
| Tinos | 1933.03 | 194,782.14 | 0.56 | 109,337.79 | 1085.08 | 0.99% |

### 3.3. Water Availability per Capita

The effectiveness of RWH application is directly related to the population benefited and more specifically to the volume of water harvested per capita. Thus, the average harvested volume per capita per simulated period is calculated, presented, and compared with historical data and the BHWR, as shown in Figure 5.

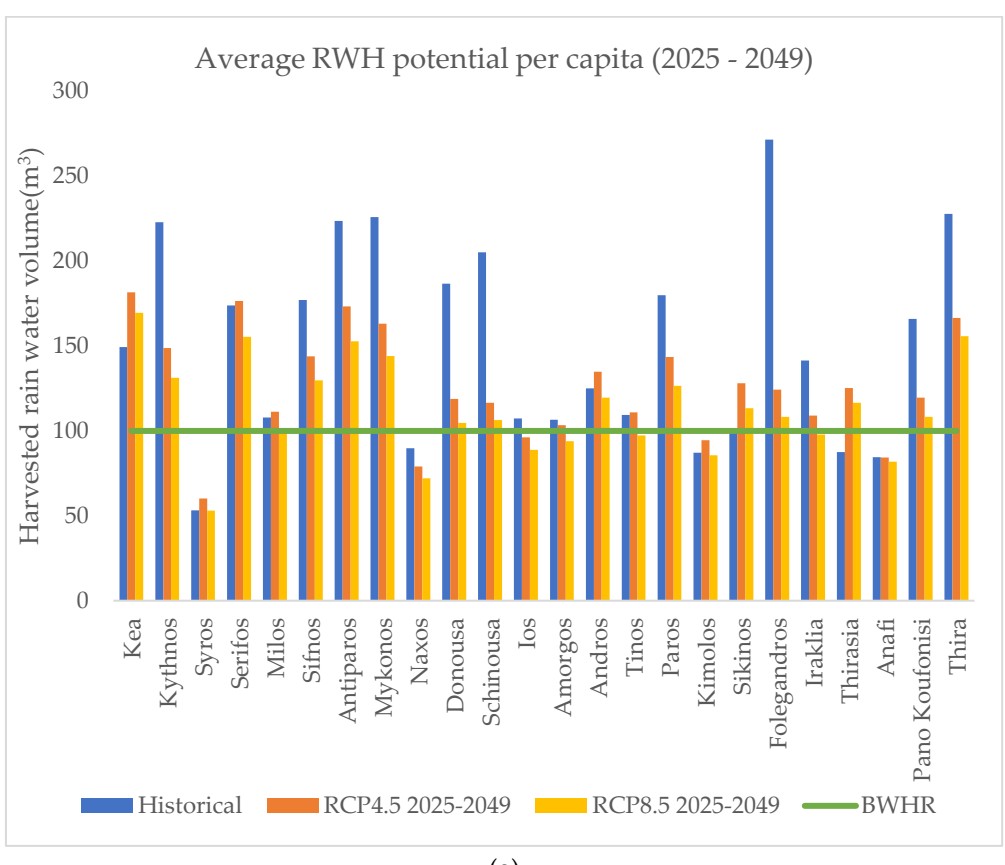

(**a**)

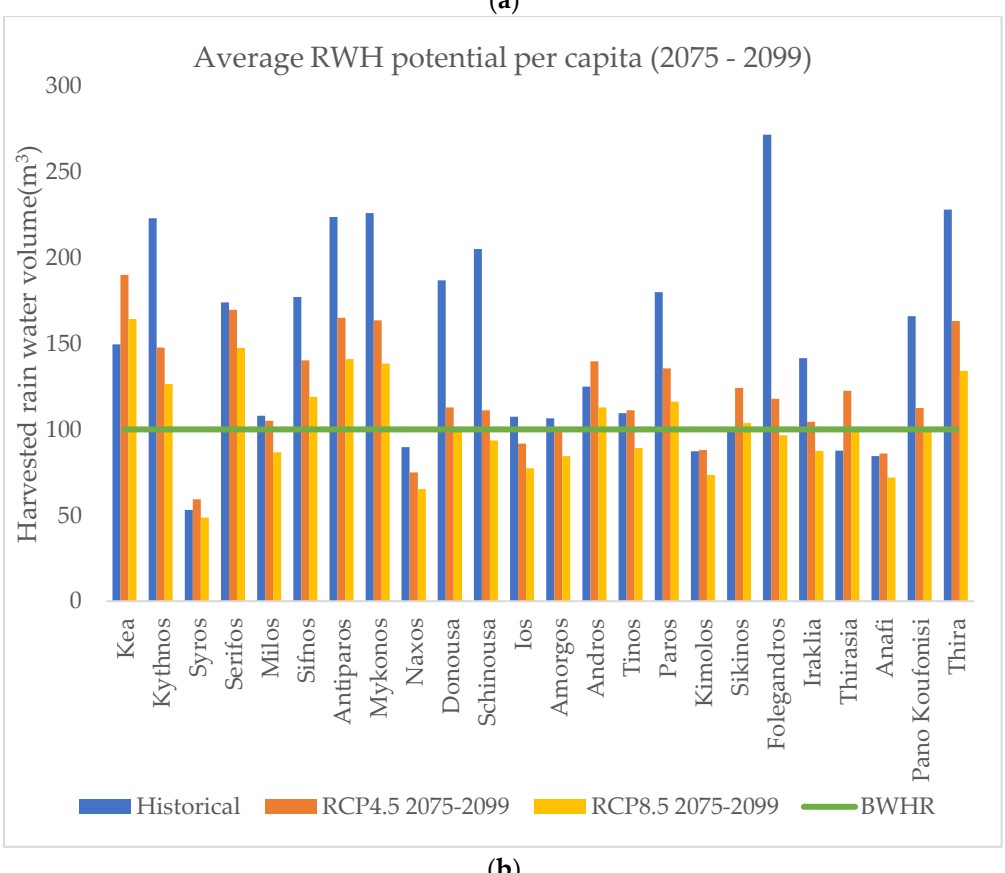

(**b**)

**Figure 5.** RWH potential per island: (**a**) 2025–2049 and (**b**) 2075–2099, based on both RCP 4.5 and RCP 8.5. These are compared with the historical data on RWH potential and the BHWR of 100 m³/p · year.

For the simulated period of 2025–2049 (Figure 5a), we observe a decrease in the available harvested water, in comparison with the historical data. This is based on the assumption that RWH applications that exist today were also available for the period of 1980–2004, since according to the Greek national land registry, the majority of constructions took place prior to 2004 in most islands. The least densely populated islands, despite the decrease in rainfall and the consequent decrease in potential harvested volume, are above the BHWR limit of 100 m$^3$/p · year. On the contrary, more populated islands (8 out of 24) fall below or are in the borderline of the BHWR limit. Thus, these islands are dependent on alternative water resources, mainly brackish and seawater desalination. It is important to note that amount of harvested water in the most populated islands of Syros and Naxos fall significantly below the threshold of 100 m$^3$/p · year; in Syros, the average water harvested is around 50 m$^3$/p · year. In the case of Sikinos and Anafi, despite the small population, the available RWH applications are limited, thus the low performance.

Similar trends can be observed for the simulated period of 2075–2099 (Figure 5b). Most islands have been reported to have a threshold below 100 m$^3$/p · year, thus remaining below the desired limit, with the addition of Iraklia for the RCP 8.5 scenario. In general, the decrease in RWH potential is observed in both the simulated scenarios, leading to the conclusion that this source of water will decrease in the future. Figures A1–A5 present a more detailed analysis of the RWH potential for the study area based on minimum, average, and maximum RWH per period.

### 3.4. Effect of Tourism

Seasonal population variation further reduces the already limited renewable water resources. This is evident in some of the most touristic Cycladic islands. ELSTAT data show that these islands received a total of 6.3 million tourists in 2021. This enormous increase in population diminishes the available renewable water resources. To calculate the effect of tourism on the RWH potential of the islands, the touristic population was normalized based on an average 5-day stay at these islands (according to ELSTAT), as shown in Table 3.

**Table 3.** Population variation on touristic Cycladic islands. The touristic activity is normalized based on an average stay of 5 days.

| Island | Local Population | Tourist Arrivals | Normalized Population |
|--------|------------------|------------------|-----------------------|
| Andros | 8883 | 276,000 | 12,664 |
| Milos | 5193 | 432,901 | 11,123 |
| Mykonos | 9802 | 1,361,196 | 28,449 |
| Naxos | 19,812 | 641,454 | 28,599 |
| Paros | 14,290 | 693,049 | 23,784 |
| Syros | 20,791 | 356,912 | 25,680 |
| Thira | 15,157 | 2,047,894 | 43,580 |
| Tinos | 8611 | 500,000 | 15,460 |

As shown in Figure 6a,b, the negative effect of seasonal population variation is significant. From Table 3, it can be seen that in the most touristic islands of Mykonos and Thira the per capita rainwater availability is reduced to 1/5 due to tourism. In contradiction to this observation, these islands harvest the largest volume of rainwater per capita (above 150 m$^3$/p · year) among all the touristic islands, with only the local population being considered. The lesser touristic islands of Andros, Milos, Paros, and Tinos borderline around the BHWR limit without the addition of tourism effect. As expected, touristic activity in these islands reduces the RWH potential by half to an average of 50 m$^3$/p · year. Finally, the most populated islands of Naxos and Syros, regardless of the touristic activity, have poor RWH potential, which is further reduced with tourism.

Comparing the effect of tourism on the RWH potential of the Cycladic islands with the different RCP scenarios, it is possible to predict the future effectiveness of this alternative water source. In general, the collectable rainwater availability shows a decreasing trend in

both RCP 4.5 and RCP 8.5 scenarios. On average, water availability per capita is decreased between 10 and 25 m³/p · year in the two scenarios. The negative effect of tourism on the Cycladic islands is evident in Figure 7.

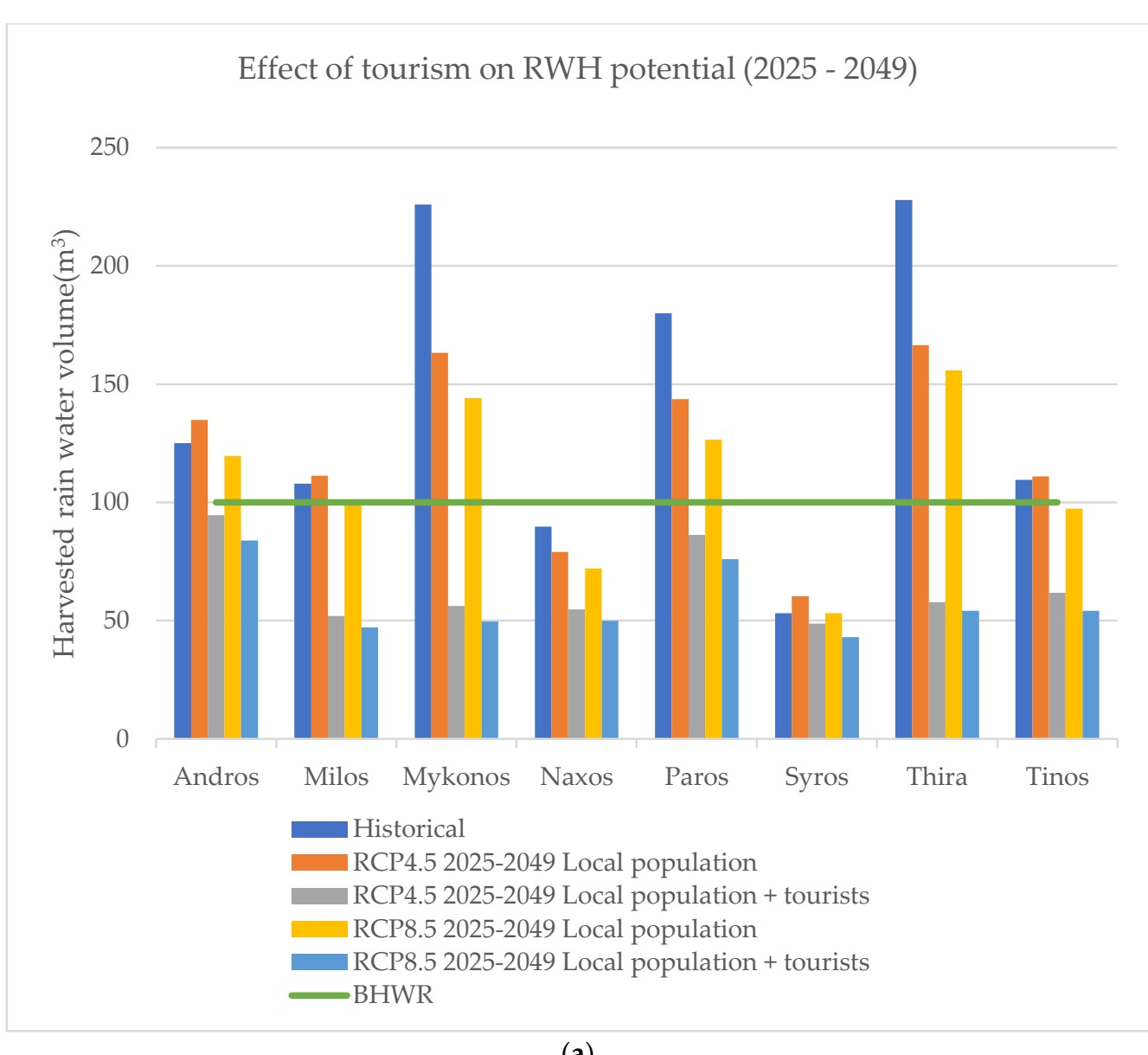

(**a**)

**Figure 6.** *Cont.*

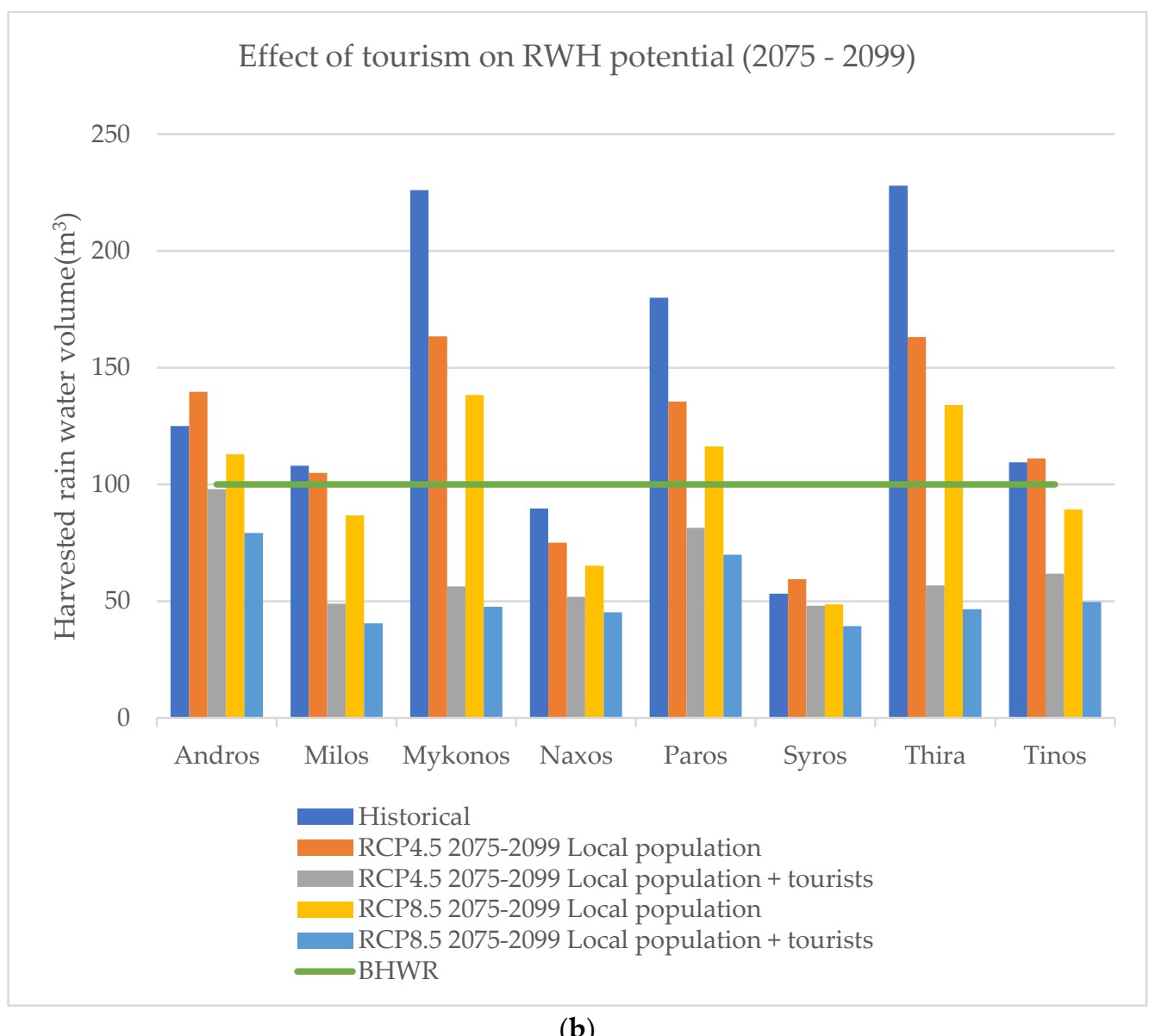

**(b)**

**Figure 6.** Effect of tourism on RWH potential per island, (**a**) 2025–2049 and (**b**) 2075–2099, based on both RCP 4.5 and RCP 8.5. These are compared with the historical RWH potential and the BHWR of 100 m$^3$/p · year.

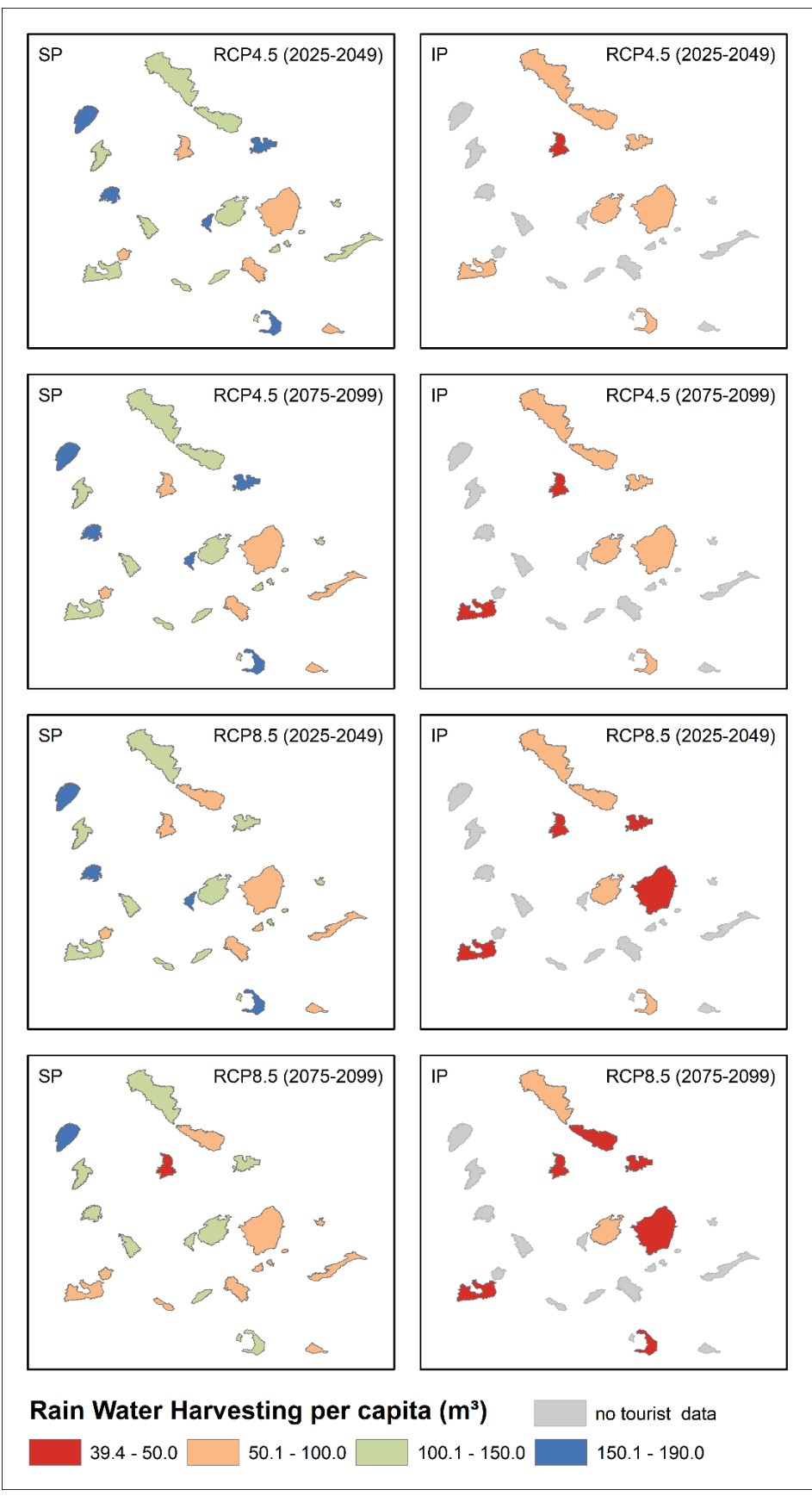

**Figure 7.** Effect of tourism on rainwater harvesting. (**Left**) Per capita RWH availability for standard population (SP). (**Right**) Per capita RWH for increased population due to touristic activity (IP).

## 4. Conclusions

Cyclades is a region that has been historically affected by water scarcity and drought, and based on high-resolution climate simulations for the next century, it can be stated that it will continue to be affected with these two phenomena. The results of this research indicate that the potential of traditional rooftop-based RWH applications, which is characteristic of the architecture of this region, will decline. The decline is based on two main factors. The first, and most evident, factor is the decrease in expected rainfall, resulting in the decrease in harvested rainwater. The second is the seasonal increase in population due to touristic activity, that increases water demands, and consequently the stress on available water resources. This is evident in the most touristic islands, where the combined effect of low rainfall and high seasonal population significantly reduces the available harvested rainwater to below the basic human water requirement of 100 m$^3$/year/capita.

To overcome this deficit in water availability, non-conventional water resources are employed, with brackish and seawater desalination being the prominent options. Brackish and seawater desalinization not only requires significant power to operate, which is associated with high carbon footprint, but it also causes the degradation of seabed flora and fauna near the water plants. Moreover, such facilities are vulnerable to a variety of external factors (power supply and raw materials spare parts availability, among others), including natural disasters (wildfires, landslides, flush floods, etc.).

Therefore, to overcome water scarcity and drought effects in the Cycladic islands, an integrated water resources management system (IWRM) should be developed. This IWRM should be based on the hierarchical use of water resources, particularly renewable water resources, with the supplementary use of brackish and seawater desalinization. Emphasis should be placed on RWH applications in both household and communities [18], given its low construction, running and maintenance costs, and proven effectiveness. RWH should be complemented with Nature-Based Solutions (NBS) for stormwater capture and storage from torrents and roads, acting as a water source for managed aquifer recharge (MAR) applications. Last but not least, wastewater treatment and reuse is another water source for MAR applications.

Along with the implementation of an IWRM to increase water availability, water-saving policies and directives must also be employed. Given the negative effect of tourism, it is important to not promote high water-consuming tourist activities (i.e., swimming pools or other high water consumption activities). Moreover, it is important to make the visitors aware of the water scarcity in the region. Additionally, advanced irrigation systems can enhance the effectiveness of irrigation, while using significantly less water, thereby decreasing the impact of agricultural activities on water resources.

To conclude, RWH applications has been playing and will continue to play a vital role as a renewable water resource to battle water scarcity and drought. However, based on climate projections for this century, RWH should be complemented with a number of alternative water resources under a well-planned IWRM system to ensure continuous water supply not only in the Cycladic islands, but in the Mediterranean region in general.

**Author Contributions:** I.Z., N.G. and S.K. analyzed and post-processed the data. N.P. conducted the climatic simulations. I.Z., N.P., N.G., S.K., D.V. and A.S. contributed equally to the preparation of this manuscript. All authors have read and agreed to the published version of the manuscript.

**Funding:** This research was funded by European Union's Horizon program "ICARIA", grant number GAP-101093806. This work was supported by computational time granted from the Greek Research and Technology Network (GRNET) in the National HPC facility, ARIS, under projects ID HRCOG (pr004020) and HRPOG (pr006028).

**Data Availability Statement:** The data presented in this study are available on request from the corresponding author.

**Conflicts of Interest:** The authors declare no conflict of interest.

**Appendix A**

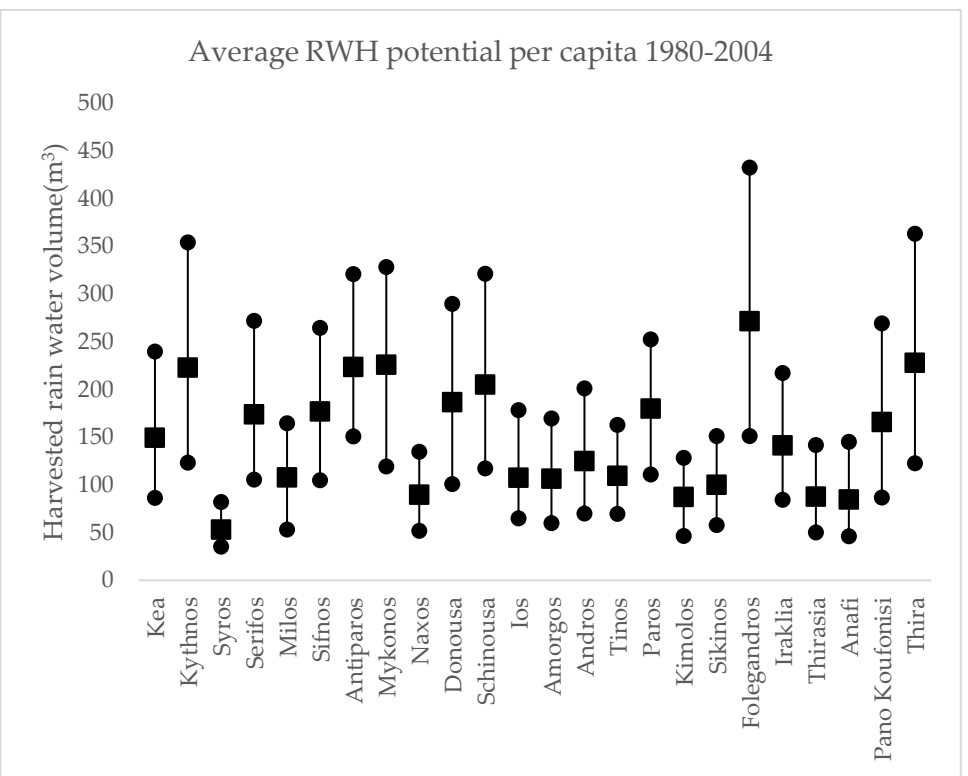

**Figure A1.** Minimum, average, and maximum RWH potential based on historical precipitation data for the period of 1980–2004.

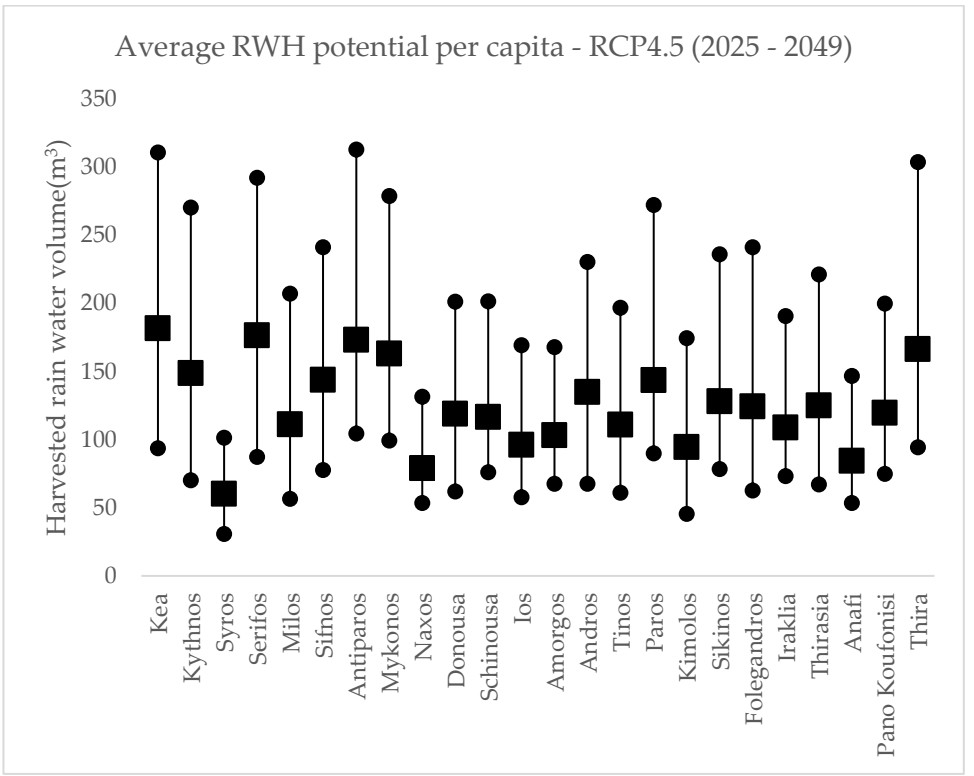

**Figure A2.** Minimum, average, and maximum RWH potential based on projected precipitation data for the period of 2025–2049 based on RCP 4.5 scenario.

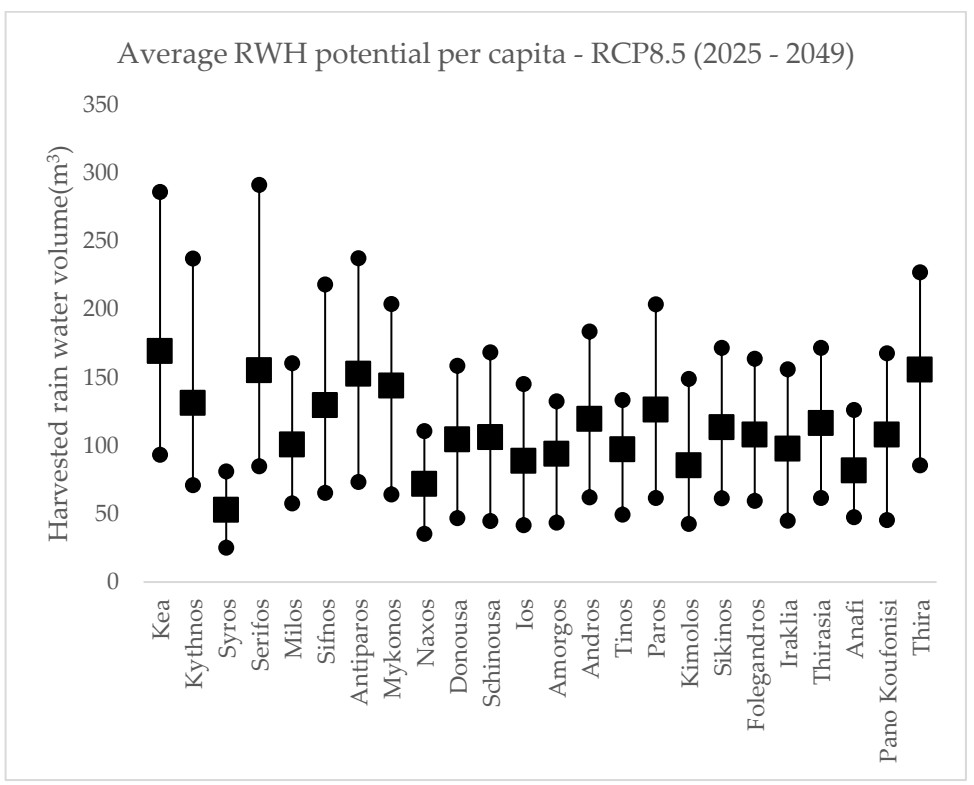

**Figure A3.** Minimum, average, and maximum RWH potential based on projected precipitation data for the period of 2025–2049 based on RCP 8.5 scenario.

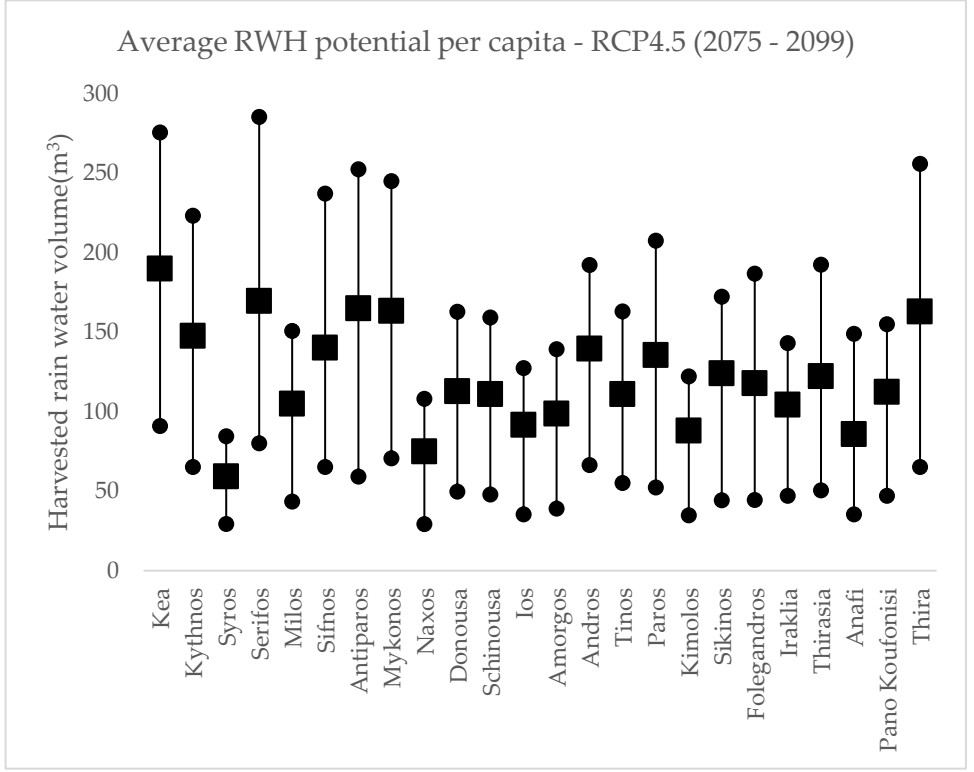

**Figure A4.** Minimum, average, and maximum RWH potential based on projected precipitation data for the period of 2075–2099 based on RCP 4.5 scenario.

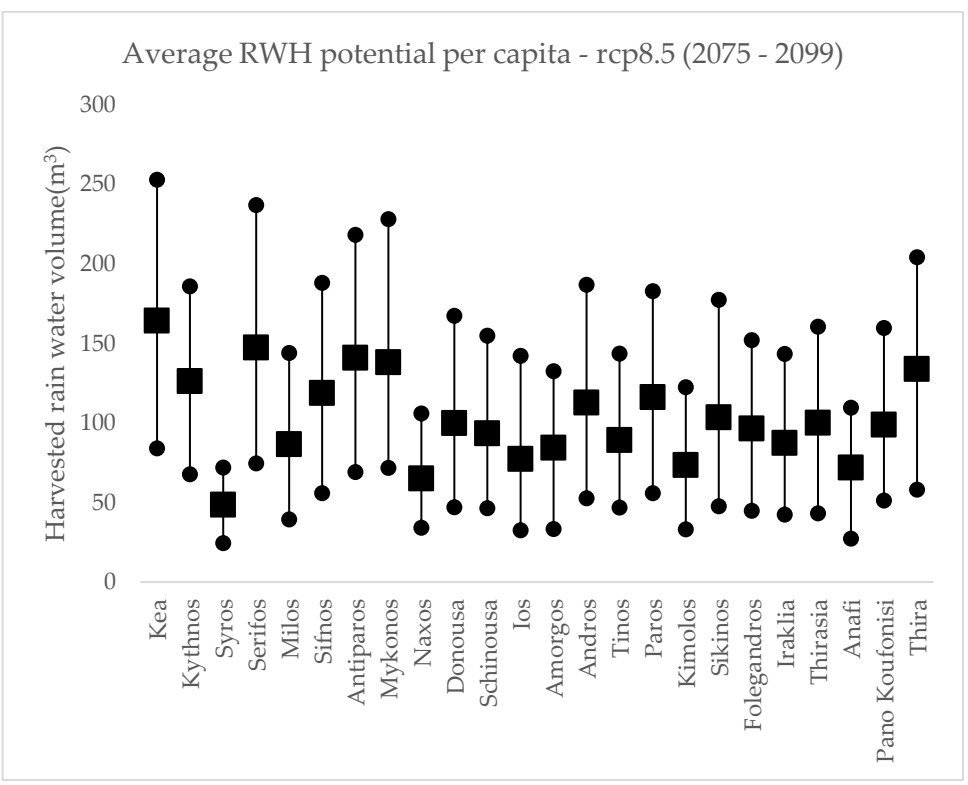

**Figure A5.** Minimum, average, and maximum RWH potential based on projected precipitation data for the period of 2075–2099 based on RCP 8.5 scenario.

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
