# Peer review of "Quantifying the Long-Term Performance of Rainwater Harvesting in Cyclades, Greece"

_water, doi:10.3390/w15173038_

Round 1
Reviewer 1 Report

English language should be revised.
Author Response
The authors would like to acknowledge the reviewer for his/her valuable time and constructive comments, which improve the quality and clarity of the paper. We would like to inform the reviewer that we have addressed all remarks and answered all their specific questions. With the requested changes, we strongly believe that our work signifies a journal-quality manuscript suitable for Water, and we appreciate their support in this direction.
Comment: Chapter 2.3 – Authors use “The historical data […] between 1980-2004”. Why recent data (2004-2023) was not considered?
Response: We would like to thank the reviewer for this question. The consideration of this specific historical period was in accordance with the availability of continuous and validated precipitation stations’ datasets, provided by the Hellenic National Meteorological Service (HNMS), for the evaluation of the downscaled model output data, analyzed in the previous chapter 2.2.
Comment: Line 216 – the indication longer than 5 months doesn’t make sense, once the scale of the figures goes from 4.1 to 6 and from 6.1 to 8
Response: We thank the reviewer for the comment. We have adapted the text to properly present all drought duration periods.
Comment: Figure 3 and 4 – include the name of the islands in the image
Response: We would like to thank the reviewer of the proposition. Figure 1 presents the islands, under investigation, by name. To ensure that Figures 3 and 4 are clear and readable for the readers, we decided not to add the named on the figures.
Comment: The analysis presented should be complemented with the maximum and minimum precipitation for the studied periods. The use of just average precipitation makes is not enough for a complete analysis.
Response: We would like to thank the reviewer of the comment. We used annual mean precipitation to eliminate any uncertainties. We have added the min/max RWH potential figures as supplementary materials. Given the number of islands under investigation, will not allow for the proper presentation of the results, based on the template of the journal.
Comment: Lines 265-269 – an image of the BHWR of all the islands should complement this description.
Response: We would like to thank the viewer for the comment. Basic Human Water Requirement is a fixes value of 100m3/year per capita. We believe that an image representing it will not provide any valuable information to the readers.
Reviewer 2 Report
1- The manuscript is not organised correctly. Some of its contents are unclear or not well presented
2- The climatic model is not transparent about how they are applied and the main procedures.
3- The methodology is not adequately described, the following points will guide you:
- The methodology needs to be explained in more detail, do they downscaling the GCm models or not?
- Check the calibration and validation of the climatic model,
-The main climatic parameters that have been applied must be defined.
More comments are on the attached pdf file.

Author Response
The authors would like to acknowledge the reviewer for his/her valuable time and constructive comments, which improve the quality and clarity of the paper. We would like to inform the reviewer that we have addressed all remarks and answered all their specific questions. With the requested changes, we strongly believe that our work signifies a journal-quality manuscript suitable for Water, and we appreciate their support in this direction.
Comment: Line 209 – Results own or previous work?
Response: We thank the reviewer for this comment. Drought duration and severity were estimated and analyzed in the previous work of Politi et al. 2022b, who studied the projected changes of drought characteristics over the area of Greece under the two RCPs. For the needs of this case study, the results in these figures were subset and remapped focusing only on the area of Cyclades Islands, to highlight in detail the areas prone to drought.
Reviewer 3 Report
This paper focused on the long-term performance of rainwater harvesting in Greece. Authors investigate and quantify the future performance of rainwater harvesting applications and their contribution to continuous, sustainable and climate resilient water supply, which selected RCP 4.5 and RCP8.5 scenario and encompass time slices representative of the historical or reference (1980–2004), near future (2025– 2049), and far future (2075–2099) periods. However, from the aspects of Abstract, methods, results, discussion and conclusions, there are still big problems that need to be improved.
The abstract needs to be re-written, and the current abstract seems to introduce the research background and significance, without reflecting the research methods, results and conclusions of the paper.
Are there different time periods described in the article?
Line257: For the simulated period 2025-2050 (Figure 3a)
In figure 4, 2025-2049?
What about 2075-2099? 2075-2100.
The introduction lacks an overview of the research progress of the relevant research methods in this paper.
The calculation results of the paper lack verification, and the methods of rainfall and population prediction simulated by meteorological models lack scientific description.
The legend has an error. For example, Figure 6.
The conclusion lacks quantitative description, so quantitative expression is recommended.
Moderate editing of English language required
Author Response
The authors would like to acknowledge the reviewer for his/her valuable time and constructive comments, which improve the quality and clarity of the paper. We would like to inform the reviewer that we have addressed all remarks and answered all their specific questions. With the requested changes, we strongly believe that our work signifies a journal-quality manuscript suitable for Water, and we appreciate their support in this direction.
Comment: Time periods described in the manuscript.
Response: We would like to thank the reviewer for this remark. We have corrected the typos related to the time periods under investigation.
Comment: The introduction lacks an overview of the research progress of the relevant research methods in this paper.
Response: We thank the reviewer for this remark. The manuscript has been adapted accordingly.
Comment: The calculation results of the paper lack verification, and the methods of rainfall and population prediction simulated by meteorological models lack scientific description.
Response: We thank the reviewer for this remark. The manuscript has been adapted accordingly.
Comment: The conclusion lacks quantitative description, so quantitative expression is recommended.
Response: We thank the reviewer for this remark. The manuscript has been adapted accordingly.
Round 2
Author Response
We would like to thank the reviewer for the constructive comments. We have adapted the manuscript accordingly.
1) The proposed literature has been added to the manuscript
2) The novelty of the work has been described.
3) We have modified the manuscript based on your remarks.
We believe that based on your constructive comments, has been improved the manuscript significantly.

Reviewer 2 Report
I believe the manuscript has been sufficiently optimized to warrant publication in Water.
Author Response
We want to thank the reviewer for the positive comments and the the constructive comments from the first reviewing round.
Reviewer 3 Report
The author basically meets the revision requirements, but it is suggested to issue a detailed revision explanation. There are some misrepresentations of details in the paper, so it is suggested that the author check and revise the whole paper more carefully.
Moderate editing of English language required
Author Response
We would like to thank the reviewer for the positive remarks. We have addressed the comments and went through the complete manuscript to ensure no misinterpretations exist.